# Performance inflation in junior tennis: Longitudinal analysis and Bayesian forecasting of ranking thresholds, efficiency, and access equity

Michal Bozděch [ID]*

Department of Physical Education and Social Sciences, Faculty of Sports Studies, Masaryk University, Brno, Czechia

* michal.bozdech@fsps.muni.cz

## Abstract

This study examined long-term trends, systemic dynamics, and predictive trajectories of junior male tennis performance within the International Tennis Federation (ITF) World Tennis Tour Juniors between 2004 and 2024. Using a longitudinal dataset of 8,082 ranked players, percentile-based performance thresholds ($P_{90}$, $P_{75}$, $P_{50}$) were first derived from annual Total Ranking Points (TRP) distributions to represent the 90th-, 75th-, and 50th-percentile cut-offs in the ITF ranking system. These empirically determined thresholds were then used as reference criteria in binary logistic regression models to estimate corresponding performance boundaries for both TRP and Points per Event (PPE)—the latter serving as an efficiency-adjusted indicator derived from the ratio of TRP to the number of tournaments played. Exponential trend fitting and Bayesian time-series forecasting (Prophet) were subsequently applied to model long-term trajectories and project performance standards for the 2025–2029 period. Findings revealed a sustained and predominantly exponential increase in both TRP and PPE percentile thresholds, indicating that competitive attainment in junior male tennis has followed an accelerating trajectory. However, growth patterns differed in magnitude: TRP thresholds exhibited consistently steeper gradients than PPE thresholds across all percentile tiers, indicating that performance inflation appears to be more strongly associated with cumulative ranking-point expansion than with uniform increases in per-event scoring efficiency, although these indicators remain strongly correlated and should therefore be interpreted as complementary rather than independent mechanisms. Although efficiency-adjusted thresholds also demonstrated upward drift—particularly at the elite level ($P_{90}$)—the magnitude of this growth remained comparatively moderate. Geographic analysis further identified a persistent concentration of elite players and tournaments within a limited group of nations, highlighting systemic inequities in access to ranking opportunities. Forecasts suggest that percentile cut-off thresholds will continue to rise over the next five years,

**Data availability statement:** All relevant data are within the paper and its Supporting Information files. To facilitate transparency and reproducibility, a reproducible analysis script documenting the main data-processing steps, statistical modelling procedures, and forecasting workflow has been provided in the S3 File (R code script). This script represents a cleaned and structured version of the analytical workflow prepared for reproducibility purposes. In addition, the empirical threshold dataset used as the primary input for the forecasting analyses has been made available in S8 Table.

**Funding:** The author(s) received no specific funding for this work.

**Competing interests:** The author have declared that no competing interests exist.

reinforcing the need for adaptive performance strategies. For coaches, federations, parents, and other key stakeholders, sustainable athlete development will depend on balancing competition access and efficiency optimisation with adequate recovery, educational commitments, and psychosocial well-being. Collectively, these results position "performance inflation" in junior tennis as a multidimensional phenomenon shaped by structural accessibility, competitive efficiency, and temporal acceleration within the ITF ranking system.

## Introduction

The International Tennis Federation (ITF) World Tennis Tour Juniors (WTTJ) serves as the principal global framework for ranking and developing emerging tennis talent. Its year-end rankings play a pivotal role in shaping player trajectories, as high-ranked juniors are far more likely to transition into professional circuits. Longitudinal evidence shows that top 10% ranked junior boys and girls ultimately reach professional status [1,2]. As such, ITF rankings have become not only a measure of athletic achievement but also a determinant of access to developmental funding, sponsorship, and selection into elite training pathways. Junior players typically commence structured training at an early age, with the transition to professional competition generally occurring between 17 and 20 years of age [3]. Recent evidence further emphasises the importance of training onset and career timing in shaping elite trajectories. Chen et al. [4] demonstrated that the top ten male players begin structured training as early as 4.3 years of age and female players at 5.6 years, typically entering the professional system around the age of 16 and reaching the world's top ten within four to six years thereafter. Similarly, top 50 players tend to initiate systematic training earlier than those ranked 51–100, underscoring that early engagement in deliberate practice constitutes a significant developmental advantage in elite tennis. However, while entry into senior-level competition begins during late adolescence, peak athletic performance in tennis tends to be reached around the age of 24 [5,6]. This temporal gap underscores the importance of sustained development beyond the junior years. Notably, only about 10% of international-level U17/U18 players successfully progress to an equivalent senior level [7], yet participation in higher-tier international junior tournaments substantially increases this probability—up to 62.1% according to recent findings [8]. These statistics highlight both the developmental significance and the structural selectivity of the ITF junior ranking system, reinforcing the necessity to understand how ranking thresholds evolve and what systemic mechanisms facilitate or constrain athletes' progression.

ITF ranking points and percentile-based performance thresholds ($P_{90}$, $P_{75}$, and $P_{50}$) serve as essential indicators of player achievement, capturing both consistency across tournaments and peak competitive capability. However, developments within the ITF junior ecosystem suggest that the rising point thresholds required to achieve elite percentile rankings may not solely reflect genuine improvements in player performance [9]. Instead, they may be partially attributable to performance inflation—a

phenomenon in which ranking standards rise due to structural expansion of tournaments and increased point-distribution opportunities rather than uniformly higher athletic quality [4]. In junior tennis, this issue is particularly salient: the number of sanctioned WTTJ tournaments has more than tripled since 2004, expanding the overall "points pool" and complicating longitudinal comparisons of competitive attainment. Additionally, because ITF rankings are inherently shaped by tournament structure and participation frequency, they may sometimes distort perceived player ability, favouring those with greater access to competition. Incorporating complementary performance metrics—such as momentum, match efficiency, or percentile-based cut-offs—could therefore refine competitive evaluation, improve forecasting accuracy, and support more informed developmental planning [10–13].

To address this challenge, the present study adopts a dual-metric framework integrating Total Ranking Points (TRP) and Points per Event (PPE). Whereas TRP captures cumulative performance and rewards frequent participation, PPE serves as an efficiency-adjusted indicator reflecting points earned relative to the number of tournaments played. This distinction allows for a more nuanced analysis of competitive success—highlighting variation in how cumulative ranking points are accumulated across differing participation density. Previous studies in performance analytics have underscored the importance of evaluating both quantity and quality of competitive exposure when assessing athletic development [14]. By combining cumulative and efficiency-based indicators, this framework provides a comprehensive means of quantifying "performance inflation" and examining how structural and behavioural factors jointly shape ranking progression in junior tennis.

Beyond individual efficiency, the spatial organisation of competitive opportunities plays an equally critical role in shaping ranking outcomes. The current ITF tournament geography reveals a persistent imbalance, with event hosting heavily concentrated in a limited number of nations—such as the United States, France, and Spain—while large regions of Africa, Asia, and South America remain underrepresented [15]. This geographic access inequity implies that athletes from tournament-dense countries enjoy greater domestic opportunities to accumulate points, whereas those from underserved regions face significant logistical and financial barriers. Such asymmetries have implications beyond fairness; they may systematically distort developmental pathways and obscure the true relationship between performance and opportunity [9,16]. Understanding how geographic concentration interacts with cumulative and efficiency-based performance metrics thus represents a crucial step toward evaluating the equity and sustainability of the ITF junior ranking structure.

To integrate these performance and structural dimensions, the present study employs a longitudinal, data-driven design spanning 2004–2024, complemented by Bayesian time-series forecasting to project percentile-based thresholds ($P_{90}$, $P_{75}$, $P_{50}$) through 2029. Binary logistic regression was first applied to estimate annual cut-off values for both TRP and PPE, while the Bayesian *Prophet* model [17] was used to model non-linear trajectories, identify changepoints, and generate probabilistic forecasts. By combining these methods, the study aims to (1) quantify and forecast percentile-based performance thresholds for both cumulative and efficiency-adjusted metrics, (2) evaluate the age profiles of players attaining these benchmarks, (3) assess geographic access equity across the elite spectrum, and (4) determine whether longitudinal trajectories display linear stability or accelerating progression over time. Through this integrated approach, the research bridges the domains of performance analytics, forecasting, and structural equity—offering a comprehensive model for understanding how competition density, efficiency, and accessibility jointly drive the evolution of ranking standards in junior male tennis.

## Methods

### Study design & data source

This study adopted a longitudinal observational design, combining retrospective trend analysis with prospective forecasting techniques. The dataset encompassed junior male players competing in the International Tennis Federation (ITF) World Tennis Tour Juniors (WTTJ) between 2004 and 2024. Although tournaments within the WTTJ (formerly ITF Junior Circuit, with grades A-C, 1–5) are categorised into multiple tiers (J10, J30, J60, J100, J200, J300, and J500), each

awarding progressively higher-ranking points, these distinctions were not included as separate analytical factors. Instead, all events were consolidated by using aggregated year-end ranking points, as provided through publicly accessible ITF records (https://www.itftennis.com/), thereby yielding a uniform performance indicator across players. This approach was adopted to capture the overall competitive standing of each athlete rather than to evaluate outcomes within specific tournament tiers. Consequently, while acknowledging the graded structure of the WTTJ events, the present analysis focused exclusively on total end-of-year points to track long-term changes and forecast future performance thresholds within junior male tennis.

Importantly, the dataset also incorporated the count of tournament entries per player and per nation, which permitted the evaluation of access equity (i.e., geographic availability of competition opportunities) and point efficiency (Points per Event, PPE). These additional variables provided the empirical basis for assessing whether rising performance thresholds reflected genuine improvements in competitiveness or were primarily driven by expanded tournament participation and unequal access across regions.

The final dataset comprised 8,082 junior male athletes whose annual performance records were systematically retrieved from the publicly available WTTJ archives for the period 2004–2024. All data were fully anonymised prior to analysis to safeguard individual identities (S1 File). The research was conducted in strict accordance with the ethical principles of the Declaration of Helsinki. Given that the study relied solely on publicly accessible, de-identified secondary data provided by the ITF, no further institutional ethical approval was required. By situating these data within the broader WTTJ framework, the study ensures that any forecasted trends in performance thresholds accurately reflect the structure and evolution of this internationally recognised ranking system.

An auxiliary dataset was constructed comprising all ITF-sanctioned junior tournaments (https://www.itftennis.com/en/tournament-calendar/world-tennis-tour-juniors-calendar) held between 2004 and 2024, including information on year, host nation, city, tournament category (Grade A-C/1–5 and J10–J500), surface type, and status (completed or cancelled). This dataset was used to derive annual counts of tournaments and host nations, serving as contextual indicators of systemic expansion and geographic access within the ITF junior circuit (S2 File).

Several potential sources of bias should be considered. First, the dataset reflects only players included in the official ITF year-end rankings and therefore may underrepresent athletes participating in fewer tournaments or regional circuits. Second, structural changes in tournament availability, particularly during the COVID-19 pandemic, may influence year-to-year comparability of performance indicators. To mitigate this effect, pandemic-affected seasons were treated separately and excluded from forecasting analyses.

## Participants and inclusion criteria

The study population comprised all junior male players listed in the official ITF year-end rankings from 2004 to 2024. Inclusion was determined based on presence in the published rankings as of 31 December for each year within the observation period. No exclusion criteria were applied with respect to performance level; players with zero Total Ranking Points were retained in the dataset, thereby ensuring an unbiased representation of the competitive field. The dataset was restricted to male (boys) players only; data from female or mixed-gender tournaments were excluded to maintain population homogeneity and analytical clarity.

## Data preprocessing and missing data

All data were obtained from publicly available ITF records and were screened for completeness prior to analysis. Observations with missing values in the key variables (Total Ranking Points, number of tournaments played, or year of birth) were excluded from derived-variable calculations (e.g., PPE or age). Because the ITF year-end ranking lists provide standardized performance records, the proportion of missing values in the analysed variables was minimal and did not exceed 1% of observations. No imputation procedures were applied.

## Variables and operational definitions

Performance thresholds were operationalised using percentile rankings—specifically, the 90th ($P_{90}$), 75th ($P_{75}$), and 50th ($P_{50}$) percentiles—calculated annually across the study period. These percentile values represented the minimum year-end ranking points required to reach each respective performance tier within the ITF junior ranking system and served as the foundation for constructing longitudinal time series of competitive thresholds.

The unit of analysis in the present study was not the individual player but the annual distribution of performance outcomes within the ITF junior ranking system. For each calendar year (2004–2024), percentile-based thresholds ($P_{90}$, $P_{75}$, $P_{50}$) were derived from the cross-sectional distribution of year-end performance values among all ranked players. These thresholds therefore represent aggregated performance benchmarks describing the competitive structure of the ranking system in a given season rather than repeated measurements of the same individuals. Consequently, the longitudinal component of the analysis refers to the temporal evolution of these aggregated percentile thresholds across years, forming six independent time series (TRP_P90, TRP_$P_{75}$, TRP_$P_{50}$, PPE_$P_{90}$, PPE_$P_{75}$, PPE_$P_{50}$) used for subsequent statistical modelling and forecasting.

In addition to Total Ranking Points (TRP), an efficiency-adjusted indicator, Points per Event (PPE), was introduced and defined as the ratio of a player's annual TRP to the total number of ITF-sanctioned tournament entries during the same calendar year; $PPE_{i,y} = TRP_{i,y}/Events_{i,y}$.

where *Events* denotes the combined number of eligible singles and doubles tournaments contested by player *i* in year *y*. This variable captures the efficiency of point accumulation per event, enabling differentiation between players achieving high totals through consistent performance and those relying primarily on participation volume.

To evaluate Geographic Access Equity, three complementary yearly metrics were computed. First, the number of nations represented by at least one player within the $P_{90}$ performance tier was calculated to indicate the breadth of international participation at the elite level. Second, the Shannon diversity index ($H'$) of these national representations was used to assess the evenness of geographic distribution, expressed both as a raw entropy measure and as its exponential form (the "effective number of countries") via the vegan package in R. This index quantifies the evenness of national representation among players in the top percentile, with higher values indicating greater geographic diversity. Third, a sensitivity criterion was applied to identify nations with at least two players within the $P_{90}$ group, providing a more conservative measure of sustained presence in the elite stratum. Together, these indicators enabled the differentiation between apparent threshold inflation—driven by the expanding number of tournaments and ranking points available—and genuine increases in competitive intensity over time. To maintain continuity of longitudinal analyses, all calendar years from 2004 to 2024 were included. However, the years 2020 and 2021 were characterised by major disruptions to the ITF junior calendar due to the COVID-19 pandemic, resulting in substantially reduced tournament availability and cross-border participation. These years were retained for transparency but interpreted as exceptional conditions rather than as indicators of systemic trends. Tournament-level data (year, host nation, event status) were analysed separately to quantify the spatial concentration of ITF Junior events. Only completed events were retained; tournaments classified as *Postponed* or *Cancelled* were excluded.

## Statistical analysis

Binary logistic regression models were employed to estimate the probability that a given performance value corresponded to each percentile threshold ($P_{90}$, $P_{75}$, $P_{50}$) within the ITF junior ranking system. In each model, the dependent variable was dichotomously coded (0, 1) to indicate whether a player's year-end performance exceeded the relevant percentile threshold. Independent predictor variables included TRP and PPE. This dual specification enabled parallel estimation of cumulative and efficiency-based performance benchmarks.

Model fit was assessed using Nagelkerke's $R^2$ as a descriptive indicator of model performance. No explicit threshold was applied for model retention; instead, the regression models were used to estimate percentile-based cut-off values for TRP and PPE across years. All models satisfying these criteria were subsequently used to derive annual cut-off

values—representing the minimum TRP and PPE required to attain the $P_{90}$, $P_{75}$, and $P_{50}$ categories. Coefficient estimates ($\hat{\beta}_0$, $\hat{\beta}_1$) from these regressions were used to compute yearly threshold points, forming the empirical basis for the time-series forecasting analyses (see supplementary material; S1 Table and S2 Table).

To evaluate whether the logistic-regression-derived threshold series introduced any material deviation from directly computed cut-offs, annual empirical percentiles ($P_{90}$, $P_{75}$, $P_{50}$) were calculated for both TRP and PPE for each year (2004–2024). These empirical threshold series were analysed using the identical forecasting configuration as the regression-based thresholds. Agreement between empirical and logistic-derived cut-offs was quantified using MAE, RMSE, maximum absolute deviation, and Pearson correlations across years. Results of this comparison are reported in the supplementary materials (S3 Table).

Annual performance thresholds were defined as empirical percentiles ($P_{90}$, $P_{75}$, $P_{50}$) of the year-end TRP and PPE distributions and constituted the primary inputs for the time-series analyses and forecasting. Binary logistic regression models were additionally fitted to characterise percentile membership as a probabilistic function of the underlying performance metric and to obtain model-based cut-offs for sensitivity comparison. Forecasting conclusions are reported for the empirically defined thresholds, with model-based thresholds presented as a robustness check in the supplementary materials (S2 and S3 Table).

Differences in points-per-event (PPE) across percentile-based performance groups (Below-$P_{50}$, $P_{50}$, $P_{75}$, $P_{90}$) were examined using the Kruskal–Wallis test, followed by Dunn's multiple comparisons with Bonferroni correction. This approach was selected due to non-normal PPE distributions and unequal variances across groups. Beyond between-group differences, the interdependence of cumulative and efficiency-based performance indicators was also examined. To further explore the relationship between cumulative and efficiency-based performance metrics, a series of correlation analyses were conducted between TRP and PPE. Spearman's rank-order correlation coefficients ($\rho$) were calculated for the entire dataset as well as separately within the percentile-defined subgroups ($P_{90}$, $P_{75}$, and $P_{50}$). This approach provided a non-parametric assessment of monotonic association strength between cumulative point accumulation and per-event efficiency. Confidence intervals for $\rho$ were estimated via bootstrap resampling (1,000 iterations), and Fisher's $z$-transformed effect sizes were reported for interpretive consistency across subsamples.

To examine potential age-related disparities across performance strata, the variable Year of Birth was used to derive players' chronological age for each observation year (Age = Year – Year of Birth). Comparative analyses were then conducted to determine whether athletes achieving the $P_{90}$, $P_{75}$, and $P_{50}$ performance thresholds differed significantly in age from their lower-ranked counterparts. Given unequal group sizes and the non-normal distribution of age data, independent-samples Mann–Whitney tests were applied to compare ranks between each percentile group (e.g., $P_{90}$ vs. non-$P_{90}$), ensuring robustness against heteroscedasticity. Descriptive statistics (mean, standard deviation, and 95% confidence intervals) and Hedges' $g$ effect sizes with 95% confidence intervals were computed for all comparisons. This procedure enabled the identification of potential maturational advantages within higher performance percentiles, providing contextual insight into the developmental profile of elite junior players.

To account for longitudinal fluctuations in competitive structure, descriptive records of the annual number of ITF-sanctioned tournaments were monitored. These data served as a contextual indicator of systemic expansion in ranking-point availability, supporting the interpretation of performance inflation patterns identified across percentile thresholds.

Further analyses were conducted to examine Geographic Access Equity, utilising two complementary metrics calculated annually: (1) the number of nations represented by at least one player within the $P_{90}$ performance tier, reflecting the breadth of international participation at the elite level; and (2) the Shannon diversity index ($H'$) of these national representations, expressed both in its raw entropy form and as its exponential equivalent—the effective number of countries—to capture the evenness of geographic distribution. In parallel, the annual count of ITF-sanctioned tournaments by host nation was examined descriptively to contextualise regional differences in event availability and potential disparities in access to ranking opportunities. Together, these indicators provided a multidimensional assessment of systemic accessibility and concentration within the upper echelon of junior male tennis.

 

## Forecasting procedures

Prior to predictive modelling, the stationarity of the percentile-based time series was assessed using the Augmented Dickey–Fuller test, which consistently indicated non-stationarity across all categories ($P_{90}$, $P_{75}$, and $P_{50}$). Given its capability to decompose non-linear and non-stationary trends, the Bayesian Prophet model was deemed appropriate for this purpose [17].

Residual temporal structure was examined through inspection of autocorrelation functions (ACF) to evaluate remaining dependence after trend extraction. The absence of statistically significant autocorrelations confirmed compliance with this assumption. Each percentile-specific time series (for both TRP and PPE) was forecasted independently using Prophet, configured to detect a maximum of 20 changepoints within the first 80% of the observed period (changepoint range = 0.8). This limit was selected to prevent overfitting and ensure interpretability of structural inflections. A Laplace prior scale ($\tau = 0.05$) was applied for changepoint regularisation, and parameter estimation was performed through Markov Chain Monte Carlo (MCMC) sampling with 2,000 iterations. Forecast uncertainty was expressed through both 80% prediction intervals and 95% Bayesian credible intervals (CI). Data corresponding to the COVID-19–affected years (2020 and 2021) were excluded from the forecasting models due to known disruptions in the global tournament schedule and player participation patterns.

To evaluate predictive robustness, rolling-origin cross-validation was conducted using sequential training windows and multi-year forecast horizons. Predictive accuracy was quantified using mean absolute error (MAE), root mean square error (RMSE), mean absolute percentage error (MAPE), and empirical coverage of credible intervals. Overall predictive performance metrics indicated satisfactory out-of-sample accuracy across all percentile series, with stable MAE and RMSE values and empirical coverage of credible intervals close to nominal levels. This validation procedure ensured that forecasts represent genuine out-of-sample predictive performance rather than simple extrapolation of historical trends.

In addition, a sensitivity analysis using log-transformed percentile series was conducted to evaluate the potential influence of distributional skewness (S4 Table); however, the primary forecasting specification was retained on the original scale to preserve the interpretability of percentile-based performance thresholds within the ITF ranking system.

Prophet was preferred over conventional ARIMA models owing to its flexibility in capturing non-linear growth patterns, its resilience to missing or irregular observations, and its suitability for complex competitive systems. Model specification and validation followed general principles recommended in the TRIPOD guidelines for predictive modelling studies, including transparent reporting of model structure, parameter estimation procedures, and out-of-sample predictive validation.

## Software and tools used

All statistical analyses—including descriptive statistics, non-parametric group comparisons, binary logistic regression, geographic equity assessment, and time-series forecasting—were conducted using JASP (Version 0.19.3, University of Amsterdam) and the R statistical environment via RStudio (Version 4.4.3, Posit Software, PBC). The *car*, *FSA*, *dplyr*, and *ggplot2* packages were used in R for non-parametric testing (Levene's test, Kruskal–Wallis test, and Dunn's post-hoc comparisons with Bonferroni adjustment), data manipulation, and visualisation. The *prophet* package was employed for Bayesian time-series forecasting, and additional diagnostic plots were generated using the *performance* library.

## Results

### Descriptive characteristics of the dataset

A total of 8,082 junior male players were recorded in the ITF year-end rankings between 2004 and 2024, encompassing 10,111 tournament entries across two decades (excluding cancelled and postponed tournaments). Descriptive statistics for Total Ranking Points (TRP), Points per Event (PPE), and tournament availability are summarised in Table 1.

Across the observation period, both TRP and PPE exhibited substantial interannual variability, indicating that the structure of the ITF junior ranking system has undergone marked fluctuations over time. The mean TRP increased from 412.22±253.81 in 2004–1,046.51±624.25 in 2024, reflecting a 154% rise over twenty years. This escalation suggests that the number of ranking points required to sustain comparable relative positions has steadily intensified, potentially signalling systemic inflation rather than purely performance-driven improvement. A comparable pattern was observed for PPE, which rose from 19.09±21.56 to 41.61±44.44 over the same period. Such a trend implies that players have become increasingly efficient at converting tournament participation into ranking points—an observation consistent with greater tournament density and point redistribution within the circuit.

Marked (mainly positive) skewness and leptokurtosis were evident throughout the dataset, with skewness values for TRP ranging from 0.56 to 6.34 and kurtosis values extending up to 58.48, indicating a persistent asymmetry dominated by a small number of exceptionally high-scoring individuals. This concentration of ranking points among a limited subset of players reinforces the view that opportunities for point accumulation have not been evenly distributed across the competitive field. A similar pattern was observed for PPE, where skewness values ranged between 1.07 and 7.57 and kurtosis values reached as high as 84.37, reflecting an even more pronounced asymmetry. These results suggest that while most players accumulated modest points per tournament, a few individuals achieved disproportionately high returns per event. Such findings point to a structural imbalance in competitive efficiency, whereby access to higher-category tournaments

**Table 1. Descriptive statistics of Total Ranking Points (TRP), Points per Event (PPE), and Tournament Availability in Junior Male Tennis (2004–2024).**

| Year | N | Total Ranking Points (TRP) | | | Points per Event (PPE) | | | Tourn. (*n*) | ΔTourn. (%) |
|---|---|---|---|---|---|---|---|---|---|
| | | M±SD | Min–Max | Skew/Kurt | M±SD | Min–Max | Skew/Kurt | | |
| 2004 | 101 | 412.2±253.8 | 90.0-2082.5 | 3.13/ 17.82 | 19.1±21.6 | 4.1-208.3 | 7.00/ 60.03 | 302 | — |
| 2005 | 88 | 445.1±255.2 | 15.0-1397.5 | 1.58/ 3.54 | 21.2±14.6 | 1.1-82.9 | 1.82/ 4.47 | 308 | +2.0 |
| 2006 | 91 | 463.4±205.4 | 116.3-1446.3 | 1.67/ 5.10 | 19.6±14.3 | 6.3-96.4 | 3.49/ 15.10 | 318 | +3.2 |
| 2007 | 104 | 424.9±222.2 | 27.5-1380 | 1.74/ 4.76 | 17.0±10.6 | 2.1-62.5 | 1.94/ 4.68 | 330 | +3.8 |
| 2008 | 97 | 437.7±190.8 | 153.8-1173.8 | 1.52/ 3.05 | 19.1±13.5 | 4.9-98 | 3.06/ 13.73 | 338 | +2.4 |
| 2009 | 90 | 455.5±194.9 | 75.0-1055.0 | 0.56/ 0.12 | 18.7±12.0 | 5.1-91.9 | 2.96/ 15.11 | 357 | +5.6 |
| 2010 | 95 | 418.8±227.1 | 85.0-1171.3 | 1.04/ 0.52 | 17.5±12.5 | 2.4-71.1 | 2.10/ 5.25 | 380 | +6.4 |
| 2011 | 96 | 462.4±238.5 | 121.3-1462.5 | 1.73/ 3.83 | 19.3±14.7 | 5.1-81.3 | 2.53/ 7.69 | 395 | +3.9 |
| 2012 | 105 | 440.8±230.3 | 110.0-1667.5 | 2.27/ 8.43 | 18.8±15.5 | 3.5-105.2 | 3.23/ 13.65 | 398 | +0.8 |
| 2013 | 100 | 482.9±232.9 | 142.5-1317.5 | 1.10/ 1.33 | 21.0±15.6 | 3.9-84.9 | 2.17/ 5.55 | 402 | +1.0 |
| 2014 | 97 | 505.7±262.4 | 108.8-1377.5 | 1.28/ 1.64 | 20.9±16.1 | 3.7-98.6 | 2.23/ 6.36 | 416 | +3.5 |
| 2015 | 104 | 471.3±227.3 | 127.5-1356.9 | 1.14/ 1.47 | 19.2±14.4 | 3.9-90.5 | 2.36/ 6.91 | 422 | +1.4 |
| 2016 | 90 | 515.4±234.4 | 130.0-1486.3 | 1.53/ 3.3 | 21.5±14.5 | 6.5-80.8 | 2.37/ 6.18 | 448 | +6.2 |
| 2017 | 103 | 482.0±216.8 | 117.5-1302.5 | 1.00/ 1.32 | 20.7±16.3 | 4.5-93.2 | 2.48/ 7.31 | 507 | +13.2 |
| 2018 | 109 | 771.4±535.2 | 56.0-3857.5 | 2.70/ 10.96 | 32.2±36.8 | 3.0-254.6 | 4.26/ 21.53 | 575 | +13.4 |
| 2019 | 102 | 742.2±523.7 | 108.8-2755.0 | 2.08/ 5.17 | 29.2±27.9 | 2.9-169.0 | 2.81/ 9.51 | 656 | +14.1 |
| 2020 | 3197 | 74.2±190.0 | 0.8-2951.3 | 6.34/ 58.48 | 3.1±7.4 | 0.0-177.6 | 9.71/ 151.65 | 274 | −58.2 |
| 2021 | 2965 | 81.4±201.7 | 0.8-2741.8 | 5.34/ 39.18 | 4.7±10.9 | 0.0-181.5 | 7.57/ 84.37 | 577 | +110.6 |
| 2022 | 114 | 933.3±476.8 | 151.3-2610.5 | 1.31/ 2.25 | 39.3±28.3 | 7.12-157.0 | 1.57/ 2.39 | 796 | +38.0 |
| 2023 | 124 | 984.9±540.0 | 124.0-2902.3 | 1.37/ 2.22 | 37.1±29.7 | 3.4-170.7 | 2.44/ 6.75 | 927 | +16.5 |
| 2024 | 110 | 1046.5±624.3 | 51.0-3393.0 | 1.38/ 2.16 | 41.6±44.4 | 2.3-261.3 | 3.33/ 11.97 | 985 | +6.3 |

**Note**: Skew＝Skewness; Kurt＝Kurtosis; *Tournaments (n)* exclude events classified as *Postponed* or *Cancelled*; only completed tournaments were included in calculating *Tournaments (n)* and Δ *Tournaments (%).*

or deeper progression within events yields an exponentially greater impact on ranking outcomes. The presence of these extreme values—particularly after the COVID-19 disruption—further accentuates the widening disparity between elite and sub-elite competitors in both total and per-event performance metrics.

Parallel shifts were observed in tournament availability. The number of completed tournaments increased from 302 in 2004–985 in 2024, representing a +226% rise. However, this expansion has not translated into a proportional broadening of access for all participants. Rather, the observed growth in tournament volume appears to have inflated the total pool of attainable points, thereby amplifying ranking dispersion. Notably, sharp fluctuations in tournament numbers were recorded during the pandemic years (2020–2021), when reductions of –58.2% in available events coincided with significant contractions in both TRP and PPE values.

To examine whether these descriptive shifts translated into corresponding changes in competitive attainment, logistic regression models were subsequently employed to identify the specific cut-off thresholds associated with the 90th, 75th, and 50th performance percentiles ($P_{90}$, $P_{75}$, and $P_{50}$). These empirically derived thresholds provided the statistical foundation for forecasting analyses, enabling the assessment of whether percentile-based standards have evolved linearly or according to an accelerating, inflationary trajectory.

**Identification of percentile-based performance thresholds**

For each analysed year between 2004 and 2024, annual performance thresholds were defined as empirical percentiles ($P_{90}$, $P_{75}$, $P_{50}$) of the year-end TRP and PPE distributions. These empirically derived cut-offs constitute the primary threshold series used for longitudinal analysis and forecasting. Binary logistic regression models were additionally fitted to characterise percentile membership probabilistically and to generate model-based cut-offs for sensitivity comparison. Separate models were estimated for Total Ranking Points (TRP) and Points per Event (PPE), thereby capturing both cumulative and efficiency-based dimensions of player performance within the ITF junior ranking system. The complete results of these regression analyses are presented in supplementary materials (S2 and S3 Table). All regression models ($n = 120$; 20 years × 3 percentiles × 2 performance indicators) were statistically significant ($p < .05$). The TRP-based models exhibited consistently strong model fit, with Nagelkerke's $R^2$ values ranging from .63 to .99. In contrast, the PPE-based models showed greater variability in in model fit ($R^2 = .14–.76$). A single exception was identified in the 2009 $P_{90}$ PPE model, which—despite reaching statistical significance ($p = .03$)—yielded a comparatively low $R^2$ (.14), suggesting weaker model fit in that specific year relative to other PPE-based models.

To illustrate the interpretive logic of the probabilistic models, the 2024 results for the $P_{90}$ percentile may be considered. In the TRP-based model, the intercept ($\hat{\beta}_0$) was –36.66 and the slope ($\hat{\beta}_1$) was 0.024 ($p < .001$, $R^2 = .91$), yielding an estimated cut-off threshold of 1,527.42 total ranking points. In contrast, the corresponding PPE model produced an intercept ($\hat{\beta}_0$) of –8.35 and a slope ($\hat{\beta}_1$) of 0.109 ($p = .003$, $R^2 = .76$), resulting in a model-based cut-off value of 76.45 points per event. These model-based thresholds closely approximated the empirically observed percentile values and followed highly similar longitudinal trajectories.

When comparing the longitudinal progression of TRP-based thresholds (Fig 1, panel A) and PPE-based thresholds (Fig 1, panel B), a broadly parallel developmental pattern was observed. In both dimensions, the years 2020 and 2021—corresponding to the COVID-19 disruption—were characterised by an abrupt collapse in the required thresholds, followed by a steep rebound and subsequent inflation in the post-pandemic period. This sharp oscillation underscores the systemic sensitivity of the ITF ranking system to structural interruptions in the tournament calendar.

Differences among percentile categories further highlight this intensification. In the TRP models (Fig 1, panel A), the smallest inter-percentile gap between $P_{90}$ and $P_{75}$ was observed in 2005 ($\Delta = 22.4$ points), while the largest occurred in 2023 ($\Delta = 319.7$ points). As illustrated in Fig 1, both TRP (panel A) and PPE (panel B) exhibited a marked upward trajectory across the observed period. Specifically, the mean PPE values increased from an average of 16.9 during 2004–2017 to 28.2 across 2018–2024, excluding the COVID-affected seasons 2020 and 2021. Collectively, these findings confirm

(A)

(B)

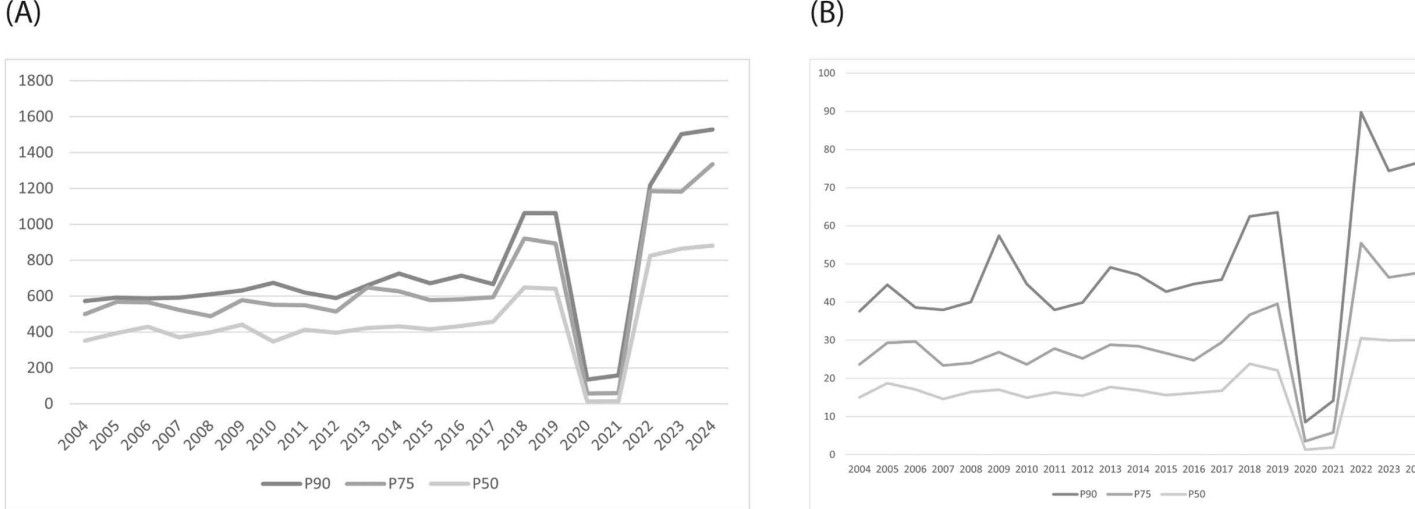

**Fig 1. Longitudinal Development of Performance Thresholds Based on Total Ranking Points (Panel A) and Points per Event (Panel B) in Junior Male Tennis (2004–2024).**

that the upper tail of performance distribution has become increasingly stratified, suggesting increased stratification within the upper tail of the distribution relative to earlier period of the ITF Junior Tour.

## Tournament participation and point-accumulation efficiency

Because the distribution of points-per-event (PPE) values was markedly non-normal within all percentile-defined performance groups (Shapiro–Wilk tests, all $p < .001$) and the homogeneity-of-variance assumption was violated (Levene's test: $F(3, 8078) = 381.71$, $p < .001$), non-parametric procedures were applied. A Kruskal–Wallis test revealed a significant effect of performance group on PPE, $\chi^2(3) = 2874.4$, $p < .001$. Subsequent pairwise comparisons using Dunn's test with Bonferroni adjustment indicated that all adjacent groups differed significantly from one another ($P_{90} > P_{75} > P_{50} >$ Below-$P_{50}$; all $p < .001$), a pattern clearly illustrated in Fig 2, which shows progressively higher PPE values and broader score dispersion toward the upper performance percentiles.

As illustrated in Fig 2, PPE values increased progressively with performance level, and the dispersion of scores broadened markedly in the upper percentiles. Descriptive statistics (Table 2) further confirmed this pattern, showing a clear, stepwise escalation in per-event scoring efficiency and tournament participation across the performance spectrum. Players outside the percentile thresholds (Below-$P_{50}$) achieved a median PPE of only 0.71 points per event (IQR = 2.81) while participating, on average, in 13.5 tournaments per year. Players meeting the $P_{50}$ criterion already displayed more intense competitive engagement (mean = 19.7 tournaments) and more than a threefold increase in efficiency (median PPE = 2.34). The $P_{75}$ group further improved both dimensions (median PPE = 5.29; mean tournaments = 24.8). The most pronounced difference was observed for the elite $P_{90}$ group, which combined the highest tournament volume (mean = 28.8 events) with the highest median efficiency (PPE = 18.9; IQR = 26.2). Taken together, these findings indicate that progression to higher percentile ranks is not achieved by tournament volume alone but by the concurrent escalation of both participation and point-accumulation efficiency.

A complementary correlation analysis was conducted to evaluate the association between cumulative performance (Total Ranking Points; TRP) and efficiency-based scoring (Points per Event; PPE). Across the full sample (N = 8082), the relationship was extremely strong ($\rho = .947$, 95% CI [.943,.949], $p < .001$), confirming that players who accumulated more

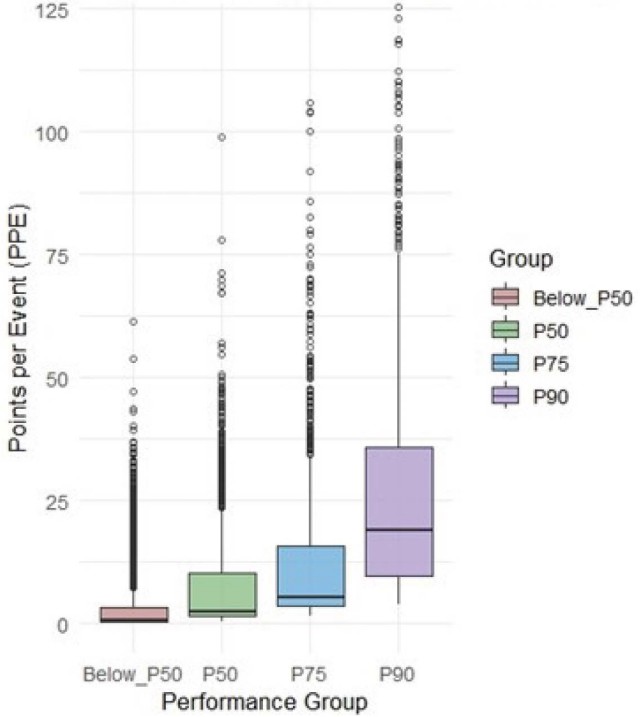

**Fig 2. Distribution of Points per Event (PPE) Across Percentile-Based Performance Groups.**

**Table 2. Descriptive Statistics of Point-Accumulation Efficiency (PPE) and Mean Tournament Participation Across Percentile-Based Performance Groups.**

| $P_i$ | $n$ | PPE | | Mean Tournaments |
|---|---|---|---|---|
| | | Median | IQR | |
| $P_{90}$ | 817 | 18.9 | 26.2 | 28.8 |
| $P_{75}$ | 1215 | 5.29 | 12.2 | 24.8 |
| $P_{50}$ | 2021 | 2.34 | 8.87 | 19.7 |
| Below_$P_{50}$ | 4029 | 0.71 | 2.81 | 13.5 |

ranking points also tended to perform more efficiently on a per-event basis. When examined within percentile-defined subgroups, the correlations remained high yet exhibited a subtle gradient: $\rho = .888$ for $P_{90}$, $\rho = .895$ for $P_{75}$, and $\rho = .903$ for $P_{50}$ (all $p < .001$). The complete correlation coefficients, confidence intervals, and Fisher's z transformations are summarised in Table 3. This pattern suggests that although total point accumulation and per-event efficiency are closely related constructs, their association weakens slightly at the elite level, where individual match outcomes and tournament selection may play a proportionally larger role. In practical terms, PPE captures an efficiency-normalised perspective on cumulative scoring that complements, rather than replaces, the cumulative perspective provided by TRP—underscoring its utility as a refined indicator of competitive performance dynamics.

Results of the Mann–Whitney U analyses (Table 4) revealed that players positioned within higher percentile tiers were consistently and significantly older than their counterparts outside these categories. All comparisons yielded highly significant differences (all $p < .001$), with effect sizes gradually decreasing from the elite to the lower performance percentiles.

**Table 3. Correlations between Total Ranking Points (TRP) and Points per Event (PPE) across the Overall Sample and Percentile-Based Performance Groups.**

| Group | n | ρ | p | 95% CI | | Fisher's z |
|---|---|---|---|---|---|---|
| | | | | Lower | Upper | |
| Total | 8082 | .947 | < .001 | .943 | .949 | 1.797 |
| - $P_{90}$ | 817 | .888 | < .001 | .868 | .906 | 1.413 |
| - $P_{75}$ | 2032 | .895 | < .001 | .882 | .906 | 1.448 |
| - $P_{50}$ | 4053 | .903 | < .001 | .895 | .911 | 1.490 |

**Table 4. Comparative Age Differences Across Performance Percentiles ($P_{90}$, $P_{75}$, and $P_{50}$) within the ITF World Tennis Tour Juniors (2004–2024).**

| $P_i$ | Age [year, M (SD)] | | p | Hedges' g | 95% CI for Hedges' g | |
|---|---|---|---|---|---|---|
| | Upper group | Lower group | | | Lower | Upper |
| $P_{90}$ | 17.3 (0.78) | 16.80 (1.12) | < .001 | .245 | .206 | .284 |
| $P_{75}$ | 17.15 (0.87) | 16.75 (1.15) | < .001 | .186 | .158 | .214 |
| $P_{50}$ | 17.03 (0.97) | 16.67 (1.20) | < .001 | .155 | .131 | .180 |

Specifically, players within the $P_{90}$ tier (top 10%) had a mean age of 17.3±0.78 years, compared with 16.80±1.12 years among the remaining cohort (Hedges' g=0.245, 95% CI [0.206, 0.284]); for the $P_{75}$ tier, the corresponding values were 17.15±0.87 and 16.75±1.15 years (g=0.186, 95% CI [0.158, 0.214]); and for the $P_{50}$ tier, 17.03±0.97 versus 16.67±1.20 years (g=0.155, 95% CI [0.131, 0.180]). These findings indicate a systematic age-related shift favouring older athletes within the WTTJ ranking structure, suggesting the presence of a relative age effect that becomes increasingly pronounced toward the upper competitive strata. This pattern reinforces the interpretation that advancement within the ITF junior system is not solely a function of performance efficiency and tournament access but is also influenced by maturational advantages that confer competitive stability within the forecasting horizon of the 2004–2029 period.

While the previous analyses highlighted that competitive success in the junior male circuit is contingent upon both the volume of tournament participation and the efficiency of point accumulation, the strong yet non-redundant correlation between TRP and PPE further underscores that these indicators capture related but distinct facets of performance. Cumulative point totals reflect sustained access and participation, whereas PPE isolates efficiency per competition. Moreover, the age-related analysis revealed that players attaining higher percentile thresholds were consistently older than their peers outside those brackets—a pattern indicative of a relative age effect within the WTTJ ranking structure. This maturational advantage, which becomes progressively more pronounced toward the top performance tiers, likely interacts with competitive exposure and access to tournaments, reinforcing disparities in developmental opportunities. However, these intertwined dynamics cannot be fully understood in isolation from the broader structural context in which they occur. Access to sanctioned tournaments varies considerably across geographic regions, potentially shaping athletes' opportunities to accrue ranking points and progress through percentile-based thresholds. Accordingly, the subsequent section examines the dimension of Geographic Access Equity, focusing on the spatial distribution of tournament availability and the representation of nations within the upper performance percentiles.

## Geographic access equity

The analysis of national representation within the upper performance percentiles revealed a pronounced geographic concentration of elite junior male players. Across the 2004–2024 period, a total of 137 nations were represented in the ITF year-end rankings; however, only a limited subset consistently produced athletes within the top $P_{90}$ performance bracket.

The top five contributing nations were the United States ($n=98$, of 728 total players), France ($n=62$, of 372 total players), Russia ($n=34$, of 296 total players), Argentina ($n=30$, of 157 total players), Italy ($n=30$, of 272 total players) and Czechia ($n=30$, of 158 total players), collectively accounting for more than one-third of all $P_{90}$-ranked players; more precisely 34.8% (Fig 3, panel A). The dominance of North American and Western European countries underscores the continued regional imbalance in the developmental pathways leading to elite junior performance.

Complementary inspection of the ITF Junior tournament calendar (S2 File) showed that this competitive dominance was only distantly related to the spatial distribution of event hosting. When postponed and cancelled events were removed, the United States ($n=367$), Mexico ($n=234$), India ($n=217$), China, P.R. ($n=213$), and Spain ($n=212$), emerged as the top 5 most frequent host nations across the 2004–2024 period. By contrast, a large group of countries—particularly from Sub-Saharan Africa, parts of Asia, and Europe, or smaller American federations—hosted only sporadic events or none at all. This asymmetric tournament geography implies that players from tournament-dense nations were able to accumulate ranking points domestically, whereas athletes from under-served regions were structurally dependent on international travel to remain competitively visible. In other words, the concentration of $P_{90}$ players in a small cluster of countries did not arise solely from superior athlete quality but was at least partly enabled by greater local opportunities to enter point-bearing events.

The number of countries represented in the $P_{90}$ category varied from as few as six (in 2009) to a maximum of twelve (in 2021), indicating fluctuations but no sustained global expansion in elite player diversity. The temporal trend in geographic diversity, expressed using the Shannon diversity index ($H'$), ranged between 1.68 and 3.76 (M=2.26, SD=0.50) across the twenty-one-year period (Fig 4). Although minor increases in $H'$ were observed in selected seasons—particularly during the early 2020s—these shifts were transient and largely attributable to short-term expansions in tournament hosting following pandemic-related disruptions. The temporary surge in the Shannon diversity index observed in 2020–2021 ($H'$ peak=3.76) reflects the transient restructuring of tournament accessibility during the COVID-19 pandemic, when the

(A)  (B)

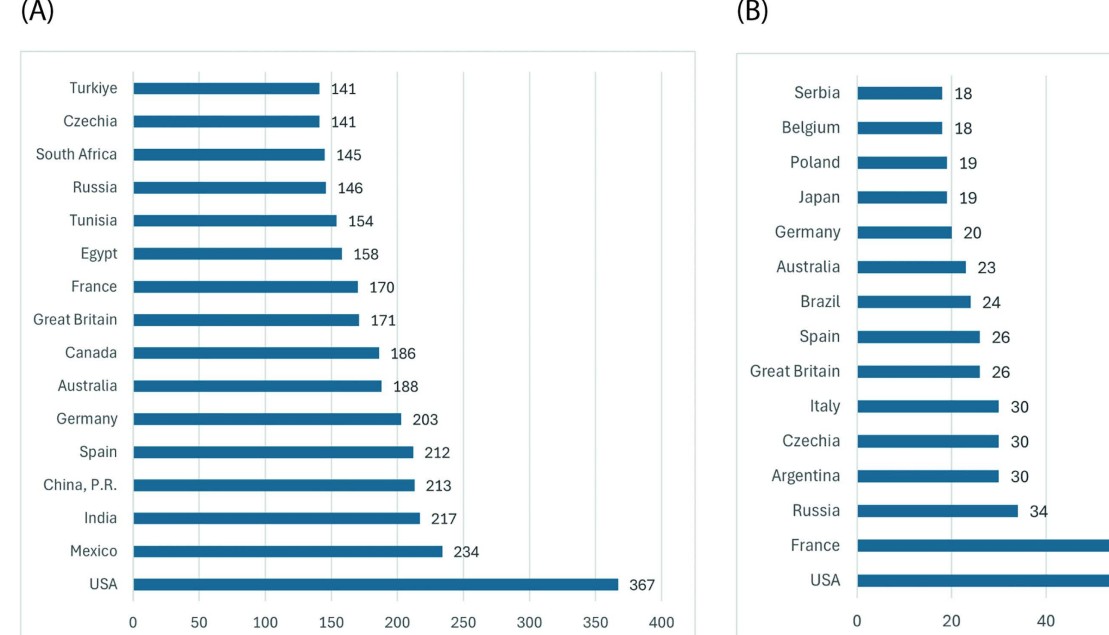

**Fig 3. Geographic Distribution of Top 15 Hosting Nations (Panel A) and $P_{90}$ Player Representation (Panel B) in the ITF World Tennis Tour Juniors, 2004–2024.**

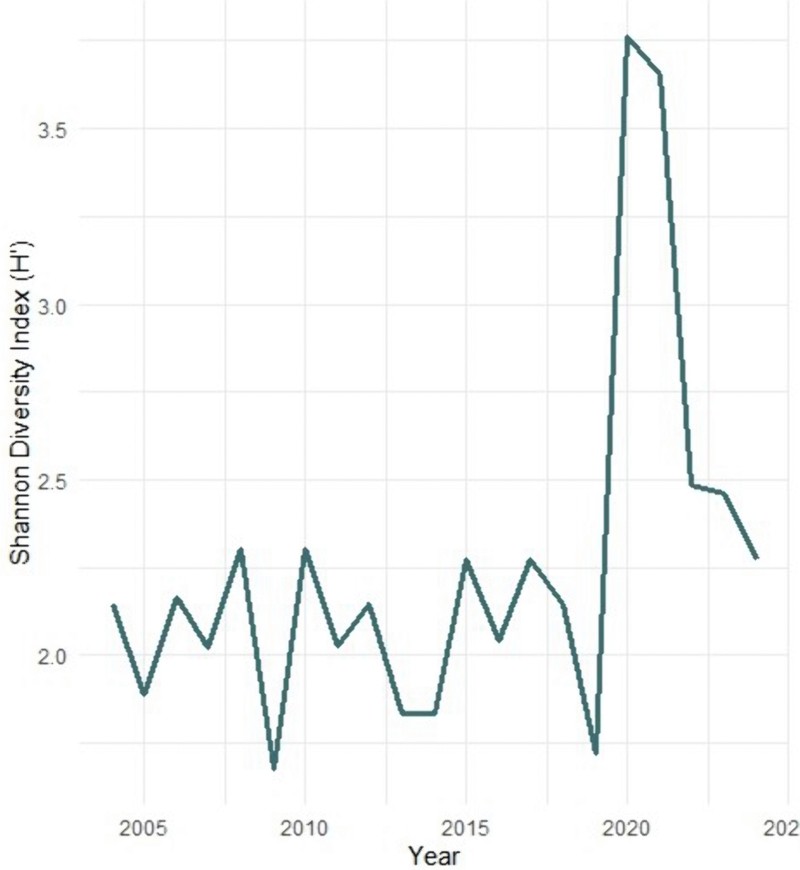

**Fig 4. Temporal Trend in Geographic Diversity among P₉₀ Players (2004–2024).**

reduction of international travel and the localisation of events momentarily increased national heterogeneity among ranked players. This anomaly did not represent a sustainable diversification of the competitive field, as diversity levels subsequently returned to pre-pandemic values by 2022.

Taken together, the findings demonstrate that the global junior tennis ecosystem remains characterised by a structural asymmetry favouring well-established tennis nations. Regions with denser tournament calendars, greater resource availability, and more mature development infrastructures continue to dominate representation within the top percentile. Consequently, the observed disparities likely reflect not only individual talent or point-accumulation efficiency but also systemic inequalities in access to international competition opportunities. The identification of these persistent geographic disparities provides critical context for the subsequent forecasting analyses, in which COVID-affected seasons will be excluded to obtain more stable projections of future performance thresholds.

The identification of persistent geographic disparities in access to international tournaments provides critical context for understanding potential future trajectories of junior tennis performance. While structural inequalities continue to shape competitive opportunities across regions, it remains essential to assess whether such imbalances are reflected in the projected evolution of performance thresholds. The following section therefore employs Bayesian time-series modelling to forecast percentile-based cut-off thresholds ($P_{90}$, $P_{75}$, and $P_{50}$) for the 2025–2029 period, offering insights into the prospective direction and pace of change within the ITF junior ranking system.

## Forecasting of future performance thresholds

To project near-term developments in competitive attainment, empirically derived percentile-based cut-off thresholds were modelled using a Bayesian Prophet specification. In accordance with the methodological framework described above, observations from the COVID-19–affected seasons (2020–2021) were excluded from model fitting in order to minimise structural distortions associated with tournament cancellations and ranking irregularities. Forecasts were generated separately for Total Ranking Points (TRP) and Points per Event (PPE), with full posterior inference obtained via Markov Chain Monte Carlo sampling (2,000 iterations). Convergence diagnostics indicated stable and well-behaved posterior distributions across all specifications (R-hat ≈ 1.00; bulk and tail effective sample sizes in the four-figure range), supporting the reliability of parameter estimation (Table 5).

As a robustness check, a sensitivity analysis was conducted in which the forecasting models were re-estimated using the full empirical percentile time series, including the COVID-19–affected seasons (2020–2021), under an identical Prophet specification. The inclusion of pandemic years resulted in systematically lower projected thresholds for the 2025–2029 horizon across all percentile categories and both performance indicators (TRP and PPE), reflecting the structural downward shock associated with tournament disruptions during the pandemic period. Rolling-origin cross-validation further indicated reduced out-of-sample predictive stability when COVID years were incorporated, supporting their treatment as a temporary structural anomaly rather than as representative long-term trend information. Detailed forecast comparisons and predictive error metrics from this sensitivity analysis are provided in the supplementary materials (S5 and S6 Table).

A log-transformation sensitivity analysis was additionally conducted to assess the extent to which forecast magnitudes depend on variance-stabilising transformations under strong skewness. Prophet models were re-estimated on log-transformed percentile series and the resulting point forecasts were back-transformed to the original scale for direct

**Table 5. Posterior Summary Statistics of Forecasting Model Parameters for Cut-Off Thresholds Based on Total Ranking Points (TRP) and Points per Event (PPE).**

| Parameter | Variable | $P_i$ | Mean | SD | 95% CI | | R-hat | ESS (bulk) | ESS (tail) |
|---|---|---|---|---|---|---|---|---|---|
| | | | | | Lower | Upper | | | |
| Growth rate (k) | TRP | $P_{90}$ | 0.237 | 0.173 | −0.116 | 0.582 | 1.005 | 1479.74 | 1804.99 |
| Offset (m) | TRP | $P_{90}$ | 0.504 | 0.054 | 0.394 | 0.610 | 1.0022 | 2011.13 | 2158.93 |
| Residual noise (σ_obs) | TRP | $P_{90}$ | 0.100 | 0.023 | 0.066 | 0.155 | 1.000 | 2764.21 | 2180.87 |
| Growth rate (k) | TRP | $P_{75}$ | 0.210 | 0.170 | −0.114 | 0.547 | 1.002 | 1774.49 | 2116.76 |
| Offset (m) | TRP | $P_{75}$ | 0.522 | 0.054 | 0.415 | 0.629 | 1.001 | 2284.36 | 2515.19 |
| Residual noise (σ_obs) | TRP | $P_{75}$ | 0.104 | 0.024 | 0.067 | 0.160 | 1.001 | 2799.13 | 2379.50 |
| Growth rate (k) | TRP | $P_{50}$ | 0.198 | 0.160 | −0.142 | 0.531 | 1.002 | 1538.53 | 1749.40 |
| Offset (m) | TRP | $P_{50}$ | 0.537 | 0.052 | 0.430 | 0.637 | 1.000 | 2251.14 | 2418.32 |
| Residual noise (σ_obs) | TRP | $P_{50}$ | 0.095 | 0.022 | 0.062 | 0.149 | 1.000 | 2425.87 | 2452.05 |
| Growth rate (k) | PPE | $P_{90}$ | 0.187 | 0.180 | −0.173 | 0.533 | 1.002 | 1643.35 | 1924.70 |
| Offset (m) | PPE | $P_{90}$ | 0.482 | 0.053 | 0.375 | 0.583 | 1.002 | 2124.92 | 2339.29 |
| Residual noise (σ_obs) | PPE | $P_{90}$ | 0.098 | 0.023 | 0.063 | 0.151 | 1.000 | 2574.57 | 2658.79 |
| Growth rate (k) | PPE | $P_{75}$ | 0.075 | 0.183 | −0.280 | 0.451 | 1.001 | 1698.12 | 1814.37 |
| Offset (m) | PPE | $P_{75}$ | 0.638 | 0.062 | 0.507 | 0.757 | 1.000 | 2131.79 | 2237.87 |
| Residual noise (σ_obs) | PPE | $P_{75}$ | 0.114 | 0.026 | 0.0744 | 0.180 | 1.001 | 2790.57 | 2216.51 |
| Growth rate (k) | PPE | $P_{50}$ | 0.062 | 0.167 | −0.278 | 0.383 | 1.002 | 1432.08 | 1821.76 |
| Offset (m) | PPE | $P_{50}$ | 0.642 | 0.050 | 0.541 | 0.738 | 1.001 | 2017.17 | 1868.34 |
| Residual noise (σ_obs) | PPE | $P_{50}$ | 0.090 | 0.021 | 0.059 | 0.141 | 1.000 | 2802.71 | 2055.92 |

comparison with the primary specification. Across all six series, the log-based models yielded substantially lower projected thresholds for 2025–2029 (mean differences approximately −46% to −76%, depending on series), indicating that the forecast levels are highly sensitive to the assumed scale of trend evolution. For transparency, the log-based forecast comparison is reported in the supplementary materials (S7 Table).

Across all percentiles and both performance indicators, posterior means of the growth-rate parameter (k) were positive. For TRP, posterior means ranged from 0.198 ($P_{50}$) to 0.237 ($P_{90}$), indicating a consistent upward drift in cumulative performance thresholds across the competitive spectrum. For PPE, the corresponding growth parameters were smaller and more heterogeneous, ranging from 0.062 ($P_{50}$) to 0.186 ($P_{90}$). Although the 95% credible intervals for k included zero in all specifications—reflecting the relatively short time-series length and the flexibility introduced by changepoints—the posterior mass was predominantly concentrated on positive values. These results suggest that, under status-quo structural conditions, percentile thresholds are more likely to continue increasing than to stabilise or decline, while acknowledging non-negligible uncertainty in long-term extrapolation.

Posterior means for the offset parameter (m) were strictly positive across all models. For TRP, m increased monotonically from $P_{90}$ (0.504) to $P_{50}$ (0.537), whereas PPE exhibited higher baseline offsets overall, particularly at $P_{75}$ (0.638) and $P_{50}$ (0.642). These positive baseline parameters indicate that the underlying trajectory of percentile thresholds is anchored at elevated levels, rendering substantial downward shifts improbable in the absence of systemic reforms.

Residual noise ($\sigma\_obs$) remained modest and stable across specifications. For TRP, $\sigma\_obs$ ranged from 0.095 to 0.104; for PPE, from 0.090 to 0.114. The similarity of these estimates across percentiles suggests comparable short-term variability around the estimated trend component in both cumulative and efficiency-adjusted indicators.

When comparing growth-rate coefficients across performance indicators, a structurally consistent pattern emerged. In all percentile categories, the posterior mean of k was higher for TRP than for PPE. This divergence was most pronounced at the median level ($P_{50}$), where TRP (0.198) substantially exceeded PPE (0.062). At the elite level ($P_{90}$), the gap narrowed but remained in favour of TRP (0.237 vs. 0.186). These findings indicate that the upward shift in percentile thresholds is more strongly reflected in cumulative ranking-point accumulation than in efficiency-adjusted scoring. While PPE thresholds also demonstrate a positive drift—particularly at the elite level—the magnitude of growth is comparatively attenuated, especially in the middle segment of the distribution. This asymmetry suggests that the long-term intensification of competitive standards within the ITF junior system is more consistently aligned with cumulative point expansion than with uniform improvements in per-event scoring efficiency.

The forecast trajectories derived from the empirical percentile-based time series are presented in Fig 5. for TRP-based thresholds ($P_{90}$, $P_{75}$, and $P_{50}$), while Fig 6 present the corresponding PPE-based projections. In both performance indicators, percentile thresholds are projected to continue increasing through 2029, with the steepest gradient consistently observed at $P_{90}$, followed by $P_{75}$ and $P_{50}$. This hierarchical ordering is preserved across the forecast horizon and reflects the structural stratification of the competitive distribution.

For TRP, projected growth remains pronounced across all percentiles, with visibly steeper slopes relative to PPE. The upper-tail threshold ($P_{90}$) exhibits the most accelerated trajectory, indicating continued expansion in the cumulative ranking points required to attain elite status. In contrast, PPE forecasts display more moderate gradients, particularly at the median level ($P_{50}$), where the projected slope is comparatively attenuated. Although efficiency-adjusted thresholds also demonstrate an upward drift, the magnitude of projected growth is consistently lower than that observed for cumulative ranking points.

The widening of credible intervals over the forecast horizon reflects the accumulation of predictive uncertainty inherent in additive trend models with changepoints. This behaviour is expected and does not imply structural instability; rather, it indicates increasing uncertainty as the projection extends further beyond the observed data range.

Importantly, the parallel upward trajectories in both TRP and PPE suggest that rising percentile thresholds cannot be attributed solely to short-term volatility or isolated structural breaks. However, the systematically stronger gradients

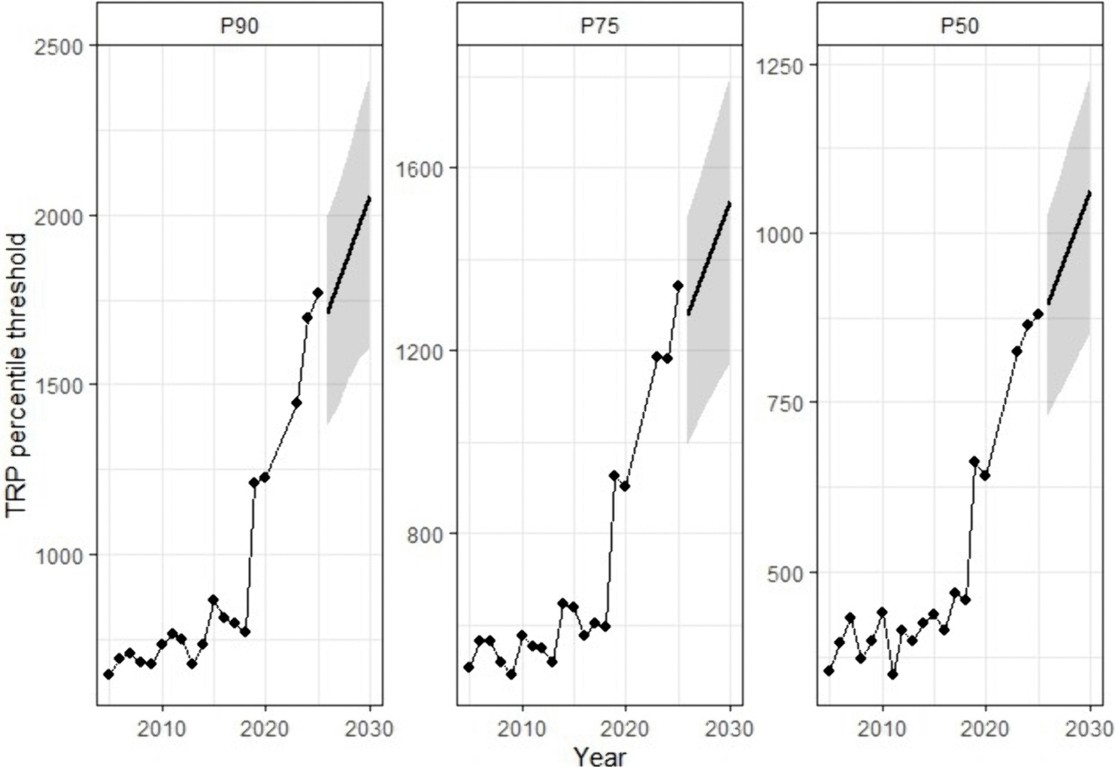

**Fig 5. Bayesian Prophet Forecasts of Cut-Off Thresholds (P$_{90}$, P$_{75}$, and P$_{50}$) for TRP.**

observed in TRP indicate that the projected inflation of competitive standards is more closely aligned with cumulative point accumulation than with uniform increases in per-event scoring efficiency. In other words, while efficiency-based thresholds continue to rise, the dominant signal of long-term intensification within the ITF junior ranking system is reflected in the expansion of total ranking points required to achieve equivalent percentile positions.

Posterior changepoint patterns indicate that most structural adjustments occurred prior to the pandemic period and were predominantly positive in direction. The persistence of upward trajectories after excluding COVID-affected seasons further supports the interpretation that the observed acceleration in percentile thresholds represents a structural, rather than transient, characteristic of the contemporary junior competitive environment.

## Discussion

Within the ITF World Tennis Tour Juniors (WTTJ), percentile-based ranking thresholds (P$_{90}$, P$_{75}$, and P$_{50}$) represent critical indicators of competitive attainment and player development. By incorporating both Total Ranking Points (TRP) and Points per Event (PPE), the present study extends the interpretive scope of these benchmarks—differentiating between cumulative point accumulation and efficiency-normalised scoring conditional on tournament exposure. This dual-parameter framework provides a more comprehensive understanding of performance inflation in junior tennis and offers practical insights for coaches, federations, and sport scientists seeking to align training, scheduling, and development strategies with the evolving competitive demands of the ITF junior system.

The analysis of percentile-based performance thresholds (P$_{90}$, P$_{75}$, and P$_{50}$) revealed a consistent and exponential upward trajectory in the ranking points required to achieve each competitive tier, confirming the accelerating intensity

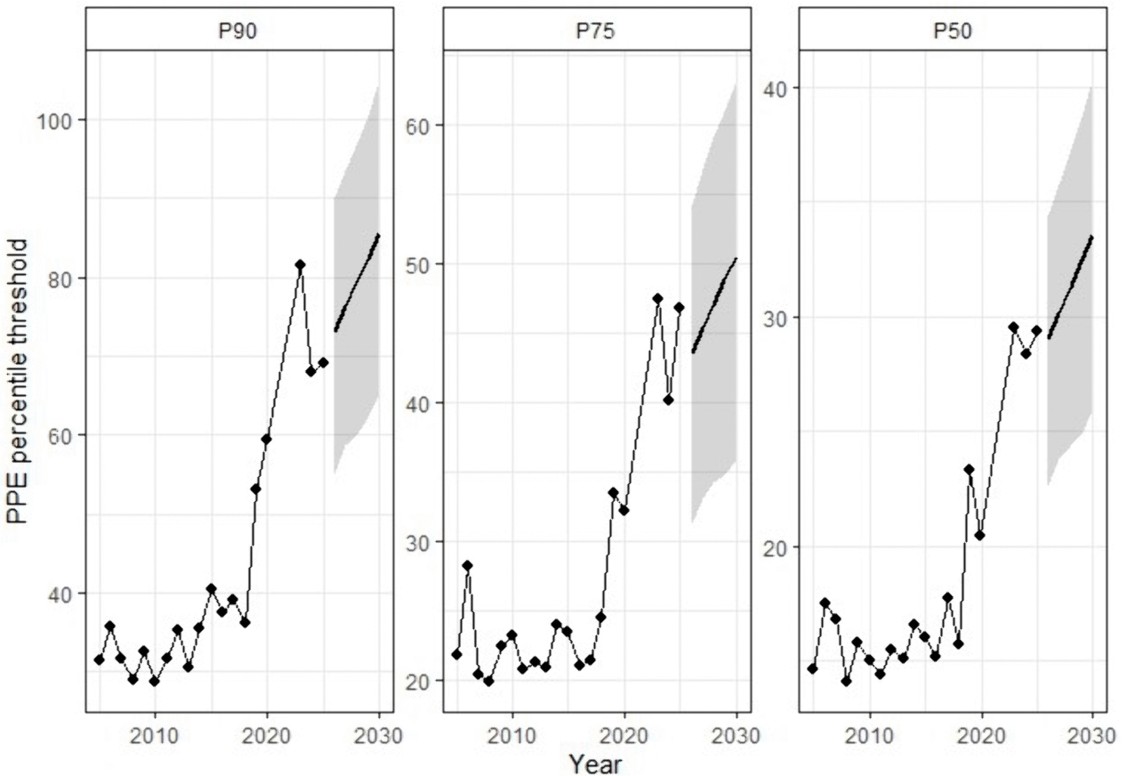

**Fig 6. Bayesian Prophet Forecasts of Cut-Off Thresholds (P$_{90}$, P$_{75}$, and P$_{50}$) PPE.**

of global junior tennis [18]. Forecasting analyses using the Bayesian Prophet model further demonstrated that these trends are expected to persist through 2029, with widening differentials between percentile categories. Such forecasts provide a valuable tool for players, coaches, and national federations: by anticipating the likely trajectory of performance benchmarks, stakeholders can set realistic performance objectives, optimise tournament scheduling, and allocate developmental resources more effectively [17,19]. When integrated into strategic planning, these forecasts help align player preparation with the projected evolution of competitive thresholds—ensuring that both cumulative (TRP) and efficiency-based (PPE) progressions are sustained within an increasingly demanding and non-stationary environment.

Because these thresholds are derived from annual cross-sectional player distributions rather than from individual longitudinal trajectories, they capture systemic changes in the competitive structure of the ITF junior ranking system rather than development pathways of specific athletes.

The present analysis demonstrated that empirically derived percentile-based thresholds provide a stable and interpretable basis for modelling competitive attainment within the ITF junior ranking structure. Annual cut-off values for TRP and PPE were computed directly from the observed distributions and subsequently modelled using Bayesian time-series forecasting to evaluate their longitudinal dynamics and projected growth. In parallel, binary logistic regression models were estimated as a complementary probabilistic framework to characterise the likelihood of exceeding specific percentile thresholds based on performance values. This modelling step did not redefine the empirical percentiles but provided an additional inferential layer for examining the relationship between cumulative and efficiency-based performance indicators. Collectively, this two-component approach—distribution-based threshold identification combined with probabilistic

modelling and forward-looking forecasting—offered a coherent analytical structure for evaluating both the structural evolution and predictive trajectory of junior male performance standards.

The consistent upward trajectory of empirically defined percentile-based cut-off thresholds observed in this study indicate an ongoing intensification of competitive standards within junior male tennis. The observed increase in percentile thresholds is consistent with the notion of performance inflation within the ITF junior ranking system. However, because cumulative ranking points (TRP) and efficiency-adjusted scoring (PPE) are mathematically and empirically closely related, the present results should be interpreted cautiously. Rather than representing fully independent mechanisms, the two indicators capture different operational perspectives on the same competitive process: cumulative accumulation of points and efficiency of point acquisition conditional on tournament exposure., suggesting that efficiency improvements contribute to, but do not dominate, threshold inflation at the elite level. This asymmetry implies that, while overall competitiveness continues to rise, sustained progression increasingly depends on optimising both tournament exposure and per-event performance efficiency through carefully planned developmental strategies. Within the context of junior tennis, tracking performance percentiles has emerged as a particularly valuable approach, as it offers a more granular assessment of a player's standing by classifying results into distinct performance tiers. This methodology provides a more comprehensive evaluation of a player's capabilities across diverse match conditions and competition levels [14].

Beyond percentile-based assessment, several complementary methods have been proposed to evaluate junior tennis performance. Physical testing protocols targeting strength, speed, and change of direction provide valuable insights into ranking potential beyond conventional rating systems such as the ITN 10.3 or USTA models [20,21]. In parallel, machine learning approaches—most notably Neural Network Auto-Regressive (NNAR) frameworks—have demonstrated superior predictive accuracy by integrating temporal and dynamic features into performance modelling [22]. While such models enhance precision and individual profiling, they remain data-intensive and technically demanding, limiting their broad application. Comparative analyses further highlight the need for robust validation and contextual integration of longitudinal data to improve predictive reliability [23]. Nevertheless, the present study demonstrates that even simplified, dual-indicator models combining TRP and PPE can effectively capture non-stationary performance dynamics and structural inequalities that more complex algorithms often overlook. This underscores the importance of developing adaptable, transparent models aligned with the accelerating and uneven evolution of the ITF junior tennis environment [24].

Beyond physical development, tournament participation frequency remains a decisive factor in ITF ranking progression. Players who compete in a higher number of sanctioned events tend to accumulate more ranking points, reflecting greater exposure to scoring opportunities and competitive adaptation [25,26]. However, findings from the present study indicate that participation volume alone does not fully explain ranking advancement. The strong positive correlation between Total Ranking Points (TRP) and Points per Event (PPE) ($\rho$ ranging from .888 to .947 across groups) demonstrates that cumulative point accumulation and per-event efficiency are closely intertwined components of competitive success. This relationship implies that while frequent participation increases scoring opportunities, the decisive differentiator among top players lies in their capacity to sustain high efficiency across tournaments. Consistent with prior research, future top 100 professionals typically engage in elite-level tournaments earlier and manage their competitive schedules strategically, achieving superior outcomes within similar participation volumes [27]. Together, these findings reinforce the joint importance of tournament access and efficiency-normalised performance efficiency in shaping progression through the ITF junior ranking system.

Although the correlation between Total Ranking Points (TRP) and Points per Event (PPE) was extremely high ($\rho$ = .947), this relationship is partly attributable to the mathematical structure of PPE, which is derived from cumulative ranking points. Consequently, PPE should not be interpreted as a statistically independent construct, but rather as an efficiency-normalised transformation of cumulative performance. Importantly, however, PPE differentiates players who achieve comparable total ranking points through differing participation densities. While TRP reflects total point accumulation, PPE captures the rate of point acquisition conditional on tournament exposure. Thus, although the two indicators are

structurally related, they represent distinct operational perspectives on competitive attainment—cumulative volume versus per-event efficiency. The present findings therefore support a complementary, rather than independent, interpretation of these performance dimensions.

The ITF junior ranking system exerts a powerful influence on both the structure and economics of global junior tennis. Attaining a high year-end position—particularly within the top 20—remains a strong predictor of later professional success, with longitudinal evidence indicating that 91% of top-ranked junior boys and 99% of junior girls progress to professional circuits [1,2]. This performance–visibility linkage incentivises national federations to expand tournament calendars and concentrate financial resources on internationally competitive players, thereby reinforcing the feedback loop between ranking attainment and access to opportunities. However, as demonstrated by the present findings, such systemic reinforcement can amplify performance inflation—in which point accumulation reflects not only athletic progression but also uneven exposure to ranking events. The geographic analysis confirmed that tournament hosting is disproportionately concentrated within a small group of nations—most prominently the United States, France, Spain, Italy, Germany, and Czechia—while large regions in Africa, Asia, and South America host only sporadic events. This spatial asymmetry in competition access structurally favours players from tournament-dense countries and magnifies the inequality between cumulative point opportunities and true competitive efficiency. Consequently, while the proliferation of tournaments has supported player development in resource-rich nations, it may simultaneously entrench structural inequities in access and developmental opportunity [15,16]. These disparities extend beyond junior competition and critically shape the transition from junior to senior levels. Consistent early participation in elite tournaments has been shown to facilitate smoother progression and enhance the likelihood of sustained professional engagement, underscoring the importance of long-term developmental programmes that provide continuous competitive exposure and institutional support [8]. The interplay between junior rankings, tournament frequency, and financial accessibility thus has direct implications for how federations allocate funding, structure international calendars, and design talent pipelines—ultimately determining the sustainability and inclusivity of future professional pathways [9].

This systemic imbalance is further evidenced by the quantitative expansion of the ITF World Tennis Tour Juniors (WTTJ), which grew from 302 events in 2004–985 in 2024—an increase of approximately 226%. Although this expansion has broadened the nominal availability of ranking opportunities, its benefits remain unevenly distributed. The majority of new tournaments have been concentrated in a limited number of host nations, reinforcing the regional asymmetry already present in player representation. For athletes from regions with sparse tournament calendars, international travel remains essential but financially prohibitive, often requiring substantial private investment and institutional support. As a result, ranking progression increasingly depends on socioeconomic background rather than sporting merit alone. While innate talent remains indispensable for long-term success, the current structure of the junior circuit rewards players who can sustain frequent international participation, as consistent tournament exposure often compensates for variability in early talent expression [28,29]. This dynamic illustrates how access inequities translate into differential developmental trajectories, underscoring the need for targeted interventions—such as travel subsidies, regional competition hubs, and scholarship programmes—to ensure that advancement within the ITF system reflects performance capability rather than resource availability.

The Bayesian Prophet model identified 14 changepoints (of max. 20 changepoints) – in the empirically derived percentile threshold series – in each percentile time series for both TRP and PPE, representing the full set of inflection points permitted under the model configuration (changepoint range = 0.8). The activation of all possible changepoints indicates a high degree of structural variability and recurrent shifts in the competitive dynamics of junior male tennis between 2004 and 2024. Temporal alignment of these changepoints revealed consistent clustering during the mid-2000s to late-2010s, preceding the COVID-19 period and reflecting sustained, pre-existing acceleration in both cumulative and efficiency-based performance thresholds. Notably, most changepoints were positive in direction, suggesting repeated episodes of growth in competitive standards rather than temporary corrections or plateaus.

These inflection points can be interpreted as structural milestones corresponding to broader transformations in the junior tennis ecosystem—such as the expansion of the ITF tournament calendar, the introduction of new ranking categories, or shifts in developmental policies emphasising early international exposure [30,31]. Integrating these events into the time-series context enhances the interpretive power of the model, enabling a more nuanced understanding of how systemic reforms, scheduling density, and talent concentration shape longitudinal performance dynamics [32,33]. Collectively, the changepoint analysis underscores that the evolution of percentile thresholds is not linear but episodic—marked by recurring surges in competitive intensity and access expansion. Such findings reinforce the need for flexible, data-driven planning within junior athlete development, where forecasting tools can be used not merely to predict future thresholds but to anticipate structural disruptions in the global competition environment [34].

## Limitations and directions for future research

While the present study offers a robust longitudinal and forecasting analysis of performance inflation within the ITF junior ranking system, several considerations warrant reflection. Forecasting models, by their probabilistic nature, describe likely trajectories rather than deterministic outcomes. Future adjustments to ITF ranking regulations, tournament structures, or developmental frameworks could therefore influence the observed trends. These potential external shifts highlight both the adaptive nature of global junior tennis and the need for continuous recalibration of predictive tools as new structural conditions emerge.

Although the exponential increase in percentile thresholds was statistically well supported, its continuation should not be presumed to be indefinite. Physiological ceilings, training saturation, and the maturing ecosystem of youth competition may gradually temper the pace of performance escalation. This limitation, however, provides a valuable opportunity for future studies to model potential saturation effects and to explore non-linear phases of competitive evolution within junior sport systems.

The use of aggregated TRP and efficiency-adjusted PPE offered a unified and methodologically transparent framework but inevitably simplified the complexity of individual player pathways. Future research could build upon this dual-metric approach by separating singles and doubles trajectories, incorporating surface preferences, or evaluating the relative weighting of event categories in determining ranking progression. Similarly, while the study identified clear geographic asymmetries in tournament distribution, future work should aim to quantify the developmental and socioeconomic impact of these disparities through econometric and accessibility modelling. Such efforts could inform evidence-based policy recommendations—such as travel subsidies, regional event clustering, or equity-focused tournament planning—to mitigate systemic imbalances in access to ranking opportunities.

The heterogeneity of the WTTJ, comprising multiple tournament tiers (J10–J500), also presents a promising avenue for refinement. The present study's decision to aggregate across these levels ensured longitudinal comparability, yet future analyses distinguishing tier-specific participation could yield insights into the stratification of developmental opportunities. Coupling this approach with hybrid statistical–machine learning techniques that integrate ranking, physical, and psychosocial indicators may further enhance both predictive precision and interpretive depth.

Although the present study primarily focused on systemic and statistical determinants of competitive attainment, future research should also integrate physiological and developmental variables to construct more holistic predictive models. Physical characteristics—such as speed, muscular strength, agility, and change-of-direction ability—constitute essential components of successful tennis performance and have been empirically linked to ranking outcomes within the ITF junior system. In particular, service velocity and upper-body strength have consistently emerged as strong predictors of ranking position among junior players [35–37]. Furthermore, early initiation of structured training and competitive participation substantially enhances the probability of attaining elite rankings, with each additional year of deliberate practice contributing meaningfully to long-term performance advancement [4,38]. Future investigations could therefore combine physiological data, developmental timing, and systemic access indicators (e.g., tournament density, travel exposure, and resource

availability) to evaluate how biological readiness interacts with structural opportunity. Such integrative, multi-level models would not only refine forecasts of junior performance trajectories but also inform evidence-based strategies for equitable and sustainable athlete development within the ITF framework.

An additional limitation concerns the strong empirical relationship between cumulative ranking points (TRP) and efficiency-adjusted scoring (PPE). Because PPE is derived from TRP and tournament participation, the two indicators are not statistically independent. Consequently, interpretations regarding distinct mechanisms underlying performance inflation should be treated as descriptive rather than causal. Future studies may benefit from incorporating additional independent performance indicators, such as match-level efficiency metrics or tournament-category weighting.

Taken together, these limitations should not be viewed as constraints but as starting points for subsequent inquiry. By extending the dual focus on cumulative and efficiency-based performance while embedding broader contextual and equity dimensions, future research can advance toward a more holistic understanding of competitive progression in junior tennis. The methodological foundation established here thus offers a flexible, scalable platform for future studies seeking to link ranking analytics, player development, and systemic accessibility within a unified empirical framework.

## Conclusion

The integrated evaluation of cumulative (TRP) and efficiency-based (PPE) indicators, together with the geographic and temporal analyses, reveals a coherent yet multidimensional pattern underlying performance inflation within the ITF junior ranking system. Taken together, the findings suggest that rising percentile thresholds are associated with both the expansion of ranking opportunities and gradual changes in efficiency-normalised scoring. However, given the strong statistical dependence between cumulative and efficiency-based indicators, these mechanisms should be interpreted as interrelated dimensions of the same competitive system rather than as fully independent drivers of performance inflation. The exponential and Bayesian time-series models applied to the empirically defined percentile thresholds have followed an accelerating trajectory, confirming that competitive attainment in junior male tennis has become progressively more demanding over the past two decades. However, the underlying forces driving this escalation differ across performance strata and are shaped by both systemic and individual determinants.

At the structural level, the continuous rise in Total Ranking Points thresholds reflects the expanding volume of sanctioned tournaments and the growing pool of available points. This systemic enlargement of the competitive environment has inflated cumulative point requirements independently of intrinsic playing quality. In contrast, the corresponding increases in PPE demonstrate that increases in efficiency-normalised scoring have also been observed—particularly within the top percentile ($P_{90}$), where the growth rate of PPE exceeded that of TRP. This divergence underscores that while participation volume remains a decisive factor for advancement in lower tiers ($P_{50}$ and $P_{75}$), elite players increasingly rely on optimised event selection and performance consistency rather than sheer tournament density. Thus, performance inflation in the ITF junior system is not a uniform artefact of structural growth but a layered phenomenon combining accessibility-driven accumulation and skill-dependent efficiency improvements.

Geographic analyses further contextualise these findings within the global structure of junior tennis. The concentration of both tournaments and elite players within a limited cluster of nations—predominantly in North America and Western Europe—illustrates how systemic access disparities contribute to the uneven distribution of competitive success. Countries hosting more tournaments not only facilitate domestic point accumulation but also enhance their athletes' probability of entering and sustaining positions within the upper percentiles. Conversely, players from underrepresented regions face structural constraints that limit their ability to translate efficiency into ranking advancement, regardless of individual capability. These geographic inequities thus amplify the systemic component of performance inflation, embedding inequality into the architecture of international junior competition.

In addition to these structural and efficiency-based mechanisms, age-related analyses revealed a consistent maturational gradient across percentile tiers, with players in higher performance brackets ($P_{90}$, $P_{75}$, $P_{50}$) being significantly older

than their peers outside those groups. This pattern indicates a relative age effect that reinforces existing performance hierarchies within the WTTJ system. Older athletes, benefiting from greater physical maturity and competitive experience, appear more likely to sustain the efficiency and consistency required for progression through elite thresholds. The presence of this maturational advantage suggests that developmental disparities in biological and chronological age constitute an additional, often overlooked, contributor to performance inflation—one that interacts with both access and efficiency dimensions.

From a forward-looking perspective, the Prophet forecasts suggest that both cumulative and efficiency-based thresholds will continue to rise through 2029, though with distinct gradients. The steeper TRP trajectories reflect persistent systemic growth, whereas the more moderate PPE increases imply incremental but meaningful improvements in player efficiency. Given the age-related asymmetries identified, future forecasting should also account for maturational and developmental timing as an integral dimension of performance modelling. In practical terms, this dual dynamic indicates that future competitive attainment will depend on a delicate balance between access and performance optimisation. For coaches, federations, parents, and other stakeholders, preparing athletes for elite progression will require not only maximising tournament participation opportunities but also refining efficiency metrics—through strategic event selection, surface specialisation, and maintaining an effective balance between singles and doubles performance—while simultaneously ensuring adequate recovery periods and supporting athletes' academic, social, and personal development. Sustainable progression within the junior circuit thus depends on coordinated planning that integrates physiological regeneration, educational commitments, and psychosocial well-being, with shared responsibility among all stakeholders to prevent overload, fatigue, and burnout in the pursuit of ranking advancement.

## Supporting information

**S1 Table. Regressions-based TRP Cut-offs.**
(CSV)

**S2 Table. Regressions-based PPE Cut-offs.**
(CSV)

**S3 Table. Empirical and Logistic-derived Cut-offs.**
(CSV)

**S4 Table. Log-transformed Percentile (Sensitivity Analysis).**
(CSV)

**S5 Table. Rolling-Origin Cross-Validation Summary for Bayesian Prophet Forecasts of Empirical Percentile Thresholds (TRP and PPE), With vs. Without Covid-Affected Years (2020–2021).**
(CSV)

**S6 Table. Five-year Forecast Comparison (2025–2029) of Empirical Percentile Thresholds.**
(CSV)

**S7 Table. Comparison of Prophet Forecasts Based on Original and Log-Transformed Percentile Series (2025–2029).**
(CSV)

**S8 Table. Forecasting Input Dataset.**
(CSV)

**S1 File. Anonymized Raw Dataset.**
(CSV)

**S2 File. Tournaments Dataset.**
(CSV)

**S3 File. Reproducible R script.**
(TXT)

## Author contributions

**Conceptualization:** Michal Bozděch.

**Data curation:** Michal Bozděch.

**Formal analysis:** Michal Bozděch.

**Methodology:** Michal Bozděch.

**Software:** Michal Bozděch.

**Supervision:** Michal Bozděch.

**Validation:** Michal Bozděch.

**Visualization:** Michal Bozděch.

**Writing – original draft:** Michal Bozděch.

**Writing – review & editing:** Michal Bozděch.

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
