## [Decision Letter · Decision Letter 0]

27 Feb 2026

PONE-D-25-61968Performance inflation in junior tennis: longitudinal analysis and Bayesian forecasting of ranking thresholds, efficiency, and access equityPLOS One

Dear Dr. Bozděch,

Thank you for submitting your manuscript to PLOS ONE. After careful consideration, we feel that it has merit but does not fully meet PLOS ONE’s publication criteria as it currently stands. Therefore, we invite you to submit a revised version of the manuscript that addresses the points raised during the review process.

**A major revision was recommended.**    **The reviewer comments are below.**

A letter that responds to each point raised by the academic editor and reviewer(s). You should upload this letter as a separate file labeled 'Response to Reviewers'.

We look forward to receiving your revised manuscript.

Kind regards,

Rachel Suet Kay Chan, Ph.D.

Academic Editor

PLOS One

Journal Requirements:

PONE-D-25-61968 – major rev./ New Academic Editor/ ARV (Straive) 27 Feb 2026:  **RTC/FTC - re-check Supporting Information files (identifying information)

SEND BACK

February 27, 2026

Journal Requirements:

https://journals.plos.org/plosone/s/file?id=wjVg/PLOSOne_formatting_sample_main_body.pdf and and and and https://journals.plos.org/plosone/s/file?id=ba62/PLOSOne_formatting_sample_title_authors_affiliations.pdf

3. We note you have included a table to which you do not refer in the text of your manuscript. Please ensure that you refer to Table 3 in your text; if accepted, production will need this reference to link the reader to the Table.

4. We note that there is identifying data in the Supporting Information file <S1 File_Dataset_WTTJ_Male_2004-2024.pdf, S1 Table_Thresholds_TRP_2004_2024, S2 File_ITF_Tournaments_2004-2024.pdf, S2 Table_Thresholds_PPE_2004_2024.pdf >. Due to the inclusion of these potentially identifying data, we have removed this file from your file inventory. Prior to sharing human research participant data, authors should consult with an ethics committee to ensure data are shared in accordance with participant consent and all applicable local laws.

-Location data

Reviewers' comments:

Reviewer's Responses to Questions

**Comments to the Author**

1. Is the manuscript technically sound, and do the data support the conclusions?

Reviewer #1: Partly

2. Has the statistical analysis been performed appropriately and rigorously? 

Reviewer #1: No

3. Have the authors made all data underlying the findings in their manuscript fully available?

The PLOS Data policy requires authors to make all data underlying the findings described in their manuscript fully available without restriction, with rare exception (please refer to the Data Availability Statement in the manuscript PDF file). The data should be provided as part of the manuscript or its supporting information, or deposited to a public repository. For example, in addition to summary statistics, the data points behind means, medians and variance measures should be available. If there are restrictions on publicly sharing data—e.g. participant privacy or use of data from a third party—those must be specified.requires authors to make all data underlying the findings described in their manuscript fully available without restriction, with rare exception (please refer to the Data Availability Statement in the manuscript PDF file). The data should be provided as part of the manuscript or its supporting information, or deposited to a public repository. For example, in addition to summary statistics, the data points behind means, medians and variance measures should be available. If there are restrictions on publicly sharing data—e.g. participant privacy or use of data from a third party—those must be specified.requires authors to make all data underlying the findings described in their manuscript fully available without restriction, with rare exception (please refer to the Data Availability Statement in the manuscript PDF file). The data should be provided as part of the manuscript or its supporting information, or deposited to a public repository. For example, in addition to summary statistics, the data points behind means, medians and variance measures should be available. If there are restrictions on publicly sharing data—e.g. participant privacy or use of data from a third party—those must be specified.requires authors to make all data underlying the findings described in their manuscript fully available without restriction, with rare exception (please refer to the Data Availability Statement in the manuscript PDF file). The data should be provided as part of the manuscript or its supporting information, or deposited to a public repository. For example, in addition to summary statistics, the data points behind means, medians and variance measures should be available. If there are restrictions on publicly sharing data—e.g. participant privacy or use of data from a third party—those must be specified.

Reviewer #1: Yes

4. Is the manuscript presented in an intelligible fashion and written in standard English?

Reviewer #1: Yes

5. Review Comments to the Author

Reviewer #1: The manuscript presents original research on longitudinal trends in junior tennis performance using a large dataset (2004-2024; n = 8082 players). The topic is relevant, and the combination of descriptive analysis, regression modeling, and forecasting is potentially valuable. The manuscript is generally well written and intelligible, and the research question is clearly stated.

However, when evaluated the requirements that “experiments, statistics, and other analyses are performed to a high technical standard” and that “conclusions are supported by the data”, there are some concerns regarding the statistical validity and reporting transparency.

Major statistical concerns

(1) Logistic regression. The use of binary logistic regression to estimate percentile thresholds is not the best choice for the goal of the paper. Percentiles are deterministic properties of the data distribution and can be computed directly. Modeling them via logistic regression introduces unnecessary complexity and potential bias without clear inferential benefit. A direct quantile-based approach or quantile regression would be more appropriate.

(2) Coherence of the statistical analysis. The workflow of the statistical analysis raises some concerns about internal coherence: Percentiles are first computed from the data, then used to define binary outcomes, and subsequently re-estimated via regression models whose outputs are later forecasted. This implies that the reported trends are, in part, artefacts of the modeling strategy rather than independent empirical findings. Why don’t use directly the empirical percentiles to make the predictions?

(3) Interpretation in the presence of collinearity. The interpretation of TRP and PPE as complementary dimensions of performance is not fully supported statistically. The extremely high correlation between them (ρ =0.947) is largely driven by their mathematical relationship, since PPE is derived from TRP. As a result, the claim that PPE captures an independent efficiency construct is not proven. This affects the validity of the conclusions.

(4) Forecasting validation. Regarding the forecasting, although the use of a Bayesian Prophet model is acceptable, the absence of any form of model validation (e.g., out-of-sample testing, cross-validation, or predictive error metrics) is a major limitation. Without validation, it is not possible to assess whether the forecasts are reliable or simply extrapolations of past trends. For instance, you could use the last years of your sample to test your model.

(5) COVID-19 pandemic years. The treatment of COVID-19 years introduces inconsistencies: the years of the pandemic (2020-2021) are excluded from forecasting but included in descriptive and regression analyses. While this decision may be reasonable, a sensitivity analysis would be necessary to demonstrate that conclusions are robust to this choice.

(6) Skewness. The variables involved in the analysis exhibit strong skewness and heavy tails (see, for example, figure 2), yet these features are not fully addressed in the modeling strategy. The absence of transformations, robustness checks, or validation of the model raises concerns about the stability of the results.

(7) Longitudinal data. The longitudinal nature of the dataset is not adequately accounted for. Observations appear to be treated as independent, despite repeated measurements across years. Instead of fitting separate models for each year, the author should be considered to fit a single model for each outcome.

(8) Interpretation. Several conclusions, particularly those regarding “performance inflation”, being driven by distinct mechanisms (structural vs efficiency-based), are not fully supported by the statistical evidence, given the strong dependence/correlation between TRP and PPE and the limitations of the modeling approach. The conclusions should therefore be presented more cautiously together with the abovementioned limitations.

Reporting and transparency (EQUATOR / PLOS ONE standards)

I strongly recommend that the author look at some of the EQUATOR network's (https://www.equator-network.org/) publishing guidelines, which, although mainly focused on the biomedical field, may help in some aspects of this article.

(9) STROBE. According to STROBE Statement E&E [1], key elements are insufficiently detailed: missing data handling, potential sources of bias, or data preprocessing steps, etc. This limits reproducibility and transparency.

(10) TRIPOD. According to TRIPOD Statement E&E [2], the manuscript falls short of expectations: There is no clear

description of model validation, no assessment of predictive performance, and limited detail on model specification.

(11) Data & Code availability. All the reporting guidelines recommend to share the data and the analysis code. The author mentions that the data is available. However, the format used to share the data (PDF in the supplementary material) is not appropriate because it does not comply with the FAIR principles (https://www.go-fair.org/fair-principles/). Furthermore, the analysis code has not been shared either. For full reproducibility, it would be desirable to provide code or more detailed information on data processing and model implementation.

Minor comments

(12) The use of an arbitrary threshold (R^2>0.20) to retain logistic models is not justified. See, for example the article of Heinze & Dunker [3].

(13) The interpretation of Nagelkerke’s R^2 as “explanatory power” should be avoided: "Power" has a specific meaning in statistics and should not be used in other contexts that could lead to ambiguity.

(14) In several parts of the manuscript, the notation of population parameters is confused with their estimation; for example, β_0,β_1,ρ should be replaced by \hat{β}_0, \hat{β}_1, \hat{ρ}.

(15) Some typos:

-P7L149. An space before “held” is missing

-P8L178 & P8L180. These two consecutive sentences are redundant.

References:

1. Vandenbroucke JP, von Elm E, Altman DG, Gøtzsche PC, Mulrow CD, Pocock SJ, Poole C, Schlesselman JJ, Egger M; STROBE Initiative. Strengthening the Reporting of Observational Studies in Epidemiology (STROBE): explanation and elaboration. Int J Surg. 2014 Dec;12(12):1500-24. doi: 10.1016/j.ijsu.2014.07.014. Epub 2014 Jul 18.

2. Collins GS, Reitsma JB, Altman DG, Moons KG. Transparent Reporting of a multivariable prediction model for Individual Prognosis or Diagnosis (TRIPOD): the TRIPOD statement. Ann Intern Med. 2015 Jan 6;162(1):55-63. doi: 10.7326/M14-0697. Erratum in: Ann Intern Med. 2015 Apr 21;162(8):600. doi: 10.7326/L15-0078-4.

3. Heinze G. and Dunkler, D. Five myths about variable selection. Transpl Int, 2017; 30: 6-10. doi: 10.1111/tri.12895

6. PLOS authors have the option to publish the peer review history of their article (what does this mean?). If published, this will include your full peer review and any attached files.). If published, this will include your full peer review and any attached files.). If published, this will include your full peer review and any attached files.). If published, this will include your full peer review and any attached files.

...

Reviewer #1: No

---

## [Author Response · Author response to Decision Letter 1]

6 Mar 2026

The opportunity to revise and resubmit the manuscript is greatly appreciated. The reviewers’ constructive and insightful comments have been carefully considered, and the manuscript has been substantially revised in response to these suggestions. All reviewer comments have been addressed in a detailed point-by-point response document. The revised manuscript has been submitted with full Track Changes in order to clearly indicate the modifications that were implemented. It is hoped that the revisions have improved the clarity, methodological transparency, and overall quality of the manuscript.

---

## [Editor Report · Decision Letter 1]

10 Mar 2026

Performance inflation in junior tennis: longitudinal analysis and Bayesian forecasting of ranking thresholds, efficiency, and access equity

PONE-D-25-61968R1

Dear Dr. Bozděch,

We’re pleased to inform you that your manuscript has been judged scientifically suitable for publication and will be formally accepted for publication once it meets all outstanding technical requirements.

Kind regards,

Rachel Suet Kay Chan, Ph.D.

Academic Editor

PLOS One
---

## [Editor Report · Acceptance letter]

PONE-D-25-61968R1

PLOS One

Dear Dr. Bozděch,

I'm pleased to inform you that your manuscript has been deemed suitable for publication in PLOS One. Congratulations! Your manuscript is now being handed over to our production team.

Kind regards,

on behalf of

Dr. Rachel Suet Kay Chan

Academic Editor

PLOS One